## [Decision Letter · Decision Letter 0]

PONE-D-24-58607Chemometric analysis of ethoxylated polymer products using extracted MALDI-TOF-MS peak distribution features.PLOS ONE

Dear Dr. Wilson,

Thank you for submitting your manuscript to PLOS ONE. After careful consideration, we feel that it has merit but does not fully meet PLOS ONE’s publication criteria as it currently stands. Therefore, we invite you to submit a revised version of the manuscript that addresses the points raised during the review process.

Although the manuscript has garnered supportive feedback from reviewers, a thorough revision is mandated to elevate its academic rigor, clarity, and overall quality, ensuring its suitability for publication.

We look forward to receiving your revised manuscript.

Kind regards,

Muhammad Atif

Academic Editor

PLOS ONE

Journal Requirements:

Additional Editor Comments:

Following peer review, the manuscript has been deemed worthy of consideration, subject to revisions that address the reviewers' comments and improve its technical merit.

Reviewers' comments:

Reviewer's Responses to Questions

**Comments to the Author**

1. Is the manuscript technically sound, and do the data support the conclusions?

Reviewer #1: Yes

Reviewer #2: Yes

Reviewer #3: Partly

2. Has the statistical analysis been performed appropriately and rigorously? 

Reviewer #1: Yes

Reviewer #2: Yes

Reviewer #3: Yes

3. Have the authors made all data underlying the findings in their manuscript fully available?

Reviewer #1: Yes

Reviewer #2: Yes

Reviewer #3: Yes

4. Is the manuscript presented in an intelligible fashion and written in standard English?

Reviewer #1: Yes

Reviewer #2: Yes

Reviewer #3: Yes

5. Review Comments to the Author

Reviewer #1: The manuscript is well written, and the authors have interpreted results very well. However, if some more latest citations are inserted, it would be more interesting. The English is of standard. The results were well interpreted.

Reviewer #2: This study developed a multivariate analysis with weighted variables based on the ratios of the resolved peak distributions, to aid in the analysis of ethoxylated products using MALDI-TOF MS. Their chemometric analysis of extracted MALDI-MS distribution features showed superior performance in identifying subtle compositional differences over and above the analysis of the full spectrum. The technical quality of the paper is assured, with a lot of data generated, and the presentation is clear. The methodology and analysis look proper and valid. Overall, the study is scientifically sound. Acceptance is recommended after minor revision.

1. In source fragmentation can occur during the ionization process in MALDI-TOF. How did the author address this issue and differentiate it from the actual analyte? I have certain doubts that the observed compounds with low EO counts were merely the ISF of high-EO ones.

2. What’s the detection limits of your targeted analyte using your methods? Is the workflow robust enough to guarantee both sensitivity and specificity good enough for correct identification? I don’t doubt whether they can be found in the manufacturer samples, just curious if it is possible to be extrapolated to other areas such as environmental samples or human specimens.

3. The author claimed that their method can aid in the accurate compound identification of ethoxylated products. I wonder if they can present some evidence as figures, such as the spectra alignment of the identified ones in comparison with the standard chemical to support the statement.

4. Were there any novel ethoxylated chemicals observed in your data mining process, which cannot be matched with any available reference? If so, it adds more value to your infrastructure as for it has the potential to identify new compounds, not only limited to the ethoxylated chemicals but can also be extrapolated to other “modificomics” analysis using MALDI-MS.

5. How will your established workflow benefit the decision-making process in a socioeconomic framework? Please briefly discuss about the reality importance and human relevance in it.

6. Line 224, “ionisation” should be “ionization”.

Reviewer #3: The paper “Chemometric analysis of ethoxylated polymer products using extracted MALDI-TOF-MS peak distribution features”, is a novel study that encompasses a new approach for the differentiation of ethoxylated products by means of an algorithm applied to the data obtained from the analysis of ethoxylated products by MALDI-TOF-MS. Thus, the work combines a large number of models for the treatment of mass spectral data, a deep linkage of high-resolution MALDI-TOF-MS analysis generating a versatile tool suitable for all users. On the other hand, I consider that the work must fulfil certain points to be considered for publication.

Línea 109: What is Croda? I see interesting to place the information about Croda, in the form: by Croda Company (London, England)

Tables 1 to 3 show similar information by changing the type of raw material. I consider it a better structure to put all the information in one table. For example: one column should be the type of ethoxylates (polysorbate 80, PEG and Oleth-20), in another column the types (1,2 and 3), and in another column the batch numbers. This proposal avoids continuous repetition of the ethoxylate name in the table title, in the table and in the table footer. In the same way, only one table footer is placed ‘for each batch of respective ethoxylates, four or five replicates were analysed by malti-tof-tof"

Tables 1 to 3: I understand that the type of ethoxylates is some variety with differences in purity, treatment, or processing time. On the other hand, it would be interesting to explain what you really mean by ‘type, 1,2, 3, R1, R1, R2, R3’.

Tables 1 to 3: Why is there talk of 4 or 5 replicas? Was it because of the technical difficulty? I would say that the samples were analysed with a minimum of 4 replicates.

Line 237: “The separation was greater for scaled data showing that smaller peaks were contributing to the difference in PS80 Type 2 samples.”

How can it be assumed that this difference is greater in the case of scaling, just by visual assessment? Although clearly type 1 is largely separated from types 2 and 3 in scaling (graph 3b), it is complex to assume only visually that types 2 and 3 have a better separation. I think it is essential to provide quantitative statistical support to show this separation. On the other hand, it is said that type 2 has a greater difference, but in the graph the greater difference is revealed for type 1 (black dots).

The same in figure 4. It is said that there is a better separation between 1 and 3.

Line 253 Fig 3: You talk about points 2 and 3 being indistinguishable, so, the above, that point 1 was better differentiated was a typo in the numerical citation?, please correct it.

As the figures have letters, do not talk about right and left graphs, talk about A and B.

Line 237: “The separation was greater for scaled data showing that smaller peaks were contributing to the difference in PS80 Type 2 samples.”

It is interesting if you are assessing the peaks of each type to be able to see these peaks.

Line 242 . “Type 2 had the lowest degree of ethoxylation with 2 to 3 EO units less than the other types.”

You said that type 2 is the one that shows the lower degree of ethoxylation, but is it really type 1 that differs. Is it really type 1 that shows the lower degree of ethoxylation? Or is it type 2, and the separation from type 1 is due to another factor that needs to be discussed?

Line 271: “This difference can also be seen in partial least squares (PLS) analysis of the extracted distribution features.”

Where? If it is figure 6 please quote it in that sentence.

As a general comment, much emphasis is made in the paper that this methodology proposed here is a useful tool for people without a deep knowledge of mass spectrometry. I understand the message, that the aim is to provide a tool that is easy to apply and versatile in the industry, which can be handled by any personnel. On the other hand, I see it unnecessary to highlight the lack of knowledge of mass spectrometry as a quality of the methodology. I would comment more on its simplicity and ease of application. I think it is possible to highlight the versatility of the methodology without the need to claim the lack of knowledge in mass spectrometry, which was necessary to create this versatile methodology. I believe that this approach is more commercially oriented than academic.

On the other hand, although I understand that it is a pioneering study in this area, the paper itself has only 16 bibliographical references, 11 of which are in the first 12 lines of the introduction. There is very little comparison of the results obtained with any other study, and there is very little discussion with other approaches.

On the other hand, the introduction is a bit short, as it gives a quick state of the art, but does not mention ethoxylates to any great extent. Thus, the chosen types polysorbate, PEG and oleth-20 are not mentioned. Applications, degree of use, general interest...

On the other hand, I congratulate the authors, the approach is attractive, and I hope it will be a tool that can be used in a common way.

6. PLOS authors have the option to publish the peer review history of their article (what does this mean? ). If published, this will include your full peer review and any attached files.

**Do you want your identity to be public for this peer review?** For information about this choice, including consent withdrawal, please see our Privacy Policy .

Reviewer #1: No

Reviewer #2: No

Reviewer #3: **Yes: ** Aly Castillo

---

## [Author Response · Author response to Decision Letter 1]

7 Mar 2025

Reviewer #1:

The manuscript is well written, and the authors have interpreted results very well. However, if some more latest citations are inserted, it would be more interesting. The English is of standard. The results were well interpreted.

We thank the reviewer for their positive comments. More recent published work has now been reviewed and added to the manuscript as citations.

Reviewer #2

This study developed a multivariate analysis with weighted variables based on the ratios of the resolved peak distributions, to aid in the analysis of ethoxylated products using MALDI-TOF MS. Their chemometric analysis of extracted MALDI-MS distribution features showed superior performance in identifying subtle compositional differences over and above the analysis of the full spectrum. The technical quality of the paper is assured, with a lot of data generated, and the presentation is clear. The methodology and analysis look proper and valid. Overall, the study is scientifically sound. Acceptance is recommended after minor revision.

We thank the reviewer for their positive comments.

1. In source fragmentation can occur during the ionization process in MALDI-TOF. How did the author address this issue and differentiate it from the actual analyte? I have certain doubts that the observed compounds with low EO counts were merely the ISF of high-EO ones.

Although MALDI-TOF is a soft ionization technique with minimal fragmentation, it is true that any fragments would occur at the same molecular weight as the low EO constituents and be difficult to differentiate. However, other analytical methods such as high-temperature GC that have been conducted on similar families of ethoxylates (outside the scope of this study) show the same Gaussian distribution of polymer lengths present in the mixture that we are observing by MALDI. We are therefore confident that the intensity of the peaks for lower EO constituents are overwhelmingly from the shorter oligomers rather than fragments of longer oligomers. The fragmentation, however, would still not impact the overall concept and methodology when making comparisons between types of polymer products or identifying and assigning the distributions to ethoxylated species in a mixture – the same ethoxylated species found in products with subtle differences in concentrations would undergo the same fragmentation, therefore the methodology described here would be unaffected, and still be applicable and useful. This comment has been added to the manuscript to make this clearer.

2. What’s the detection limits of your targeted analyte using your methods? Is the workflow robust enough to guarantee both sensitivity and specificity good enough for correct identification? I don’t doubt whether they can be found in the manufacturer samples, just curious if it is possible to be extrapolated to other areas such as environmental samples or human specimens.

The focus of this manuscript was solely to focus on the methods to study and compare industrial polymer products rather than testing their presence in biological samples, therefore we cannot comment on testing human specimens as it was outside of the scope of the study. However, from other non-published work that has been conducted by the same group to detect ethoxylated contaminants in wool grease, we believe it would be possible to extrapolate the method to these samples, to not only detect but compare and identify the type of ethoxylated compounds present, though again, this is outside the scope of this study.

3. The author claimed that their method can aid in the accurate compound identification of ethoxylated products. I wonder if they can present some evidence as figures, such as the spectra alignment of the identified ones in comparison with the standard chemical to support the statement.

The data analysed is from a calibrated MALDI-MS instrument and therefore generates accurate mass to deduce the product composition. This technique is therefore different to relative techniques such as GPC which require polymer standards for comparison to deduce any product information. We have therefore not analysed any standards and are unable to provide any figures of spectral alignment to standards, as the accuracy of our data will be superior to other relative techniques. A comment to explain this has now been added to the manuscript.

Incidentally, the products analysed are supplied by Croda Europe Ltd - Croda is a specialty chemical manufacturer that often produce the standards that are sold via third party chemical standard suppliers such as Sigma-Aldrich, therefore the ethoxylates studied in this investigation are likely to be the standards themselves.

4. Were there any novel ethoxylated chemicals observed in your data mining process, which cannot be matched with any available reference? If so, it adds more value to your infrastructure as for it has the potential to identify new compounds, not only limited to the ethoxylated chemicals but can also be extrapolated to other “modificomics” analysis using MALDI-MS.

Please see above comment regarding available references. No new compounds of unknown composition were tested, as production of these ethoxylated chemicals is tightly controlled and planned, especially as any ethoxylation reaction is so high risk. However, although specific details can’t be disclosed in this manuscript for manufacturer’s confidentiality, novel process conditions and modifications to produce new types of existing products (novel versions) with Croda’s portfolio have been investigated. A comment has been added to the manuscript to make this clearer.

5. How will your established workflow benefit the decision-making process in a socioeconomic framework? Please briefly discuss about the reality importance and human relevance in it.

Beyond industrial decision-making regarding plant levelling, quality and business reputations, these methods could be extrapolated to various applications which directly affect the wider population. The concepts described in the manuscripts could be applied to ensure pharmaceuticals or consumer products remain safe and the composition remains the same when new sustainable technologies are employed to produce them. Similarly, as described earlier, these methods can also be used to detect ethoxylated products as contaminants to other products, ensuring safety and quality to consumers. Finally, although not shown in this study, our methods have been used to find subtle differences between products made from vegetable based raw materials and those from animal origin, therefore these methods could be applied to testing products to ensure compliance for products suitable for vegetarians, or those people preferring to purchase products from sustainable sources. A comment has been added to the manuscript to make this clearer.

6. Line 224, “ionisation” should be “ionization”.

This has now been corrected in the manuscript.

Reviewer #3

The paper “Chemometric analysis of ethoxylated polymer products using extracted MALDI-TOF-MS peak distribution features”, is a novel study that encompasses a new approach for the differentiation of ethoxylated products by means of an algorithm applied to the data obtained from the analysis of ethoxylated products by MALDI-TOF-MS. Thus, the work combines a large number of models for the treatment of mass spectral data, a deep linkage of high-resolution MALDI-TOF-MS analysis generating a versatile tool suitable for all users. On the other hand, I consider that the work must fulfil certain points to be considered for publication.

1. Línea 109: What is Croda? I see interesting to place the information about Croda, in the form: by Croda Company (London, England)

This information has now been corrected and updated in the manuscript.

2. Tables 1 to 3 show similar information by changing the type of raw material. I consider it a better structure to put all the information in one table. For example: one column should be the type of ethoxylates (polysorbate 80, PEG and Oleth-20), in another column the types (1,2 and 3), and in another column the batch numbers. This proposal avoids continuous repetition of the ethoxylate name in the table title, in the table and in the table footer. In the same way, only one table footer is placed ‘for each batch of respective ethoxylates, four or five replicates were analysed by malti-tof-tof"

The tables have now been changed to the format suggested by the reviewer.

3. Tables 1 to 3: I understand that the type of ethoxylates is some variety with differences in purity, treatment, or processing time. On the other hand, it would be interesting to explain what you really mean by ‘type, 1,2, 3, R1, R1, R2, R3’.

This relates to original Table 6 in the manuscript (now changed to Table 4) of different ratios of EO to fatty alcohol. This has now been made clearer in the manuscript, linking both new Table 1 and new Table 4.

4. Tables 1 to 3: Why is there talk of 4 or 5 replicas? Was it because of the technical difficulty? I would say that the samples were analysed with a minimum of 4 replicates.

This information has now been updated in the manuscript as suggested by the reviewer.

5. Line 237: “The separation was greater for scaled data showing that smaller peaks were contributing to the difference in PS80 Type 2 samples.”

How can it be assumed that this difference is greater in the case of scaling, just by visual assessment? Although clearly type 1 is largely separated from types 2 and 3 in scaling (graph 3b), it is complex to assume only visually that types 2 and 3 have a better separation. I think it is essential to provide quantitative statistical support to show this separation. On the other hand, it is said that type 2 has a greater difference, but in the graph the greater difference is revealed for type 1 (black dots).

The same in figure 4. It is said that there is a better separation between 1 and 3.

Line 253 Fig 3: You talk about points 2 and 3 being indistinguishable, so, the above, that point 1 was better differentiated was a typo in the numerical citation?, please correct it.

This section of the results, Figure 3 and Figure 4 that the reviewer discusses have all been checked by the authors, and several mistakes were identified, where “Type 1” and “Type 2” were used interchangeably and incorrectly.

The numbering of the types has now been edited and corrected. The results now discuss Type 1 (black dots) being separate from Types 2 and 3 (indistinguishable) using the full spectrum analysis. All types are separated in Figure 4, with better separation between Types 2 and 3 using ratio-scaled extracted features. We thank the reviewer for identifying this.

6. As the figures have letters, do not talk about right and left graphs, talk about A and B.

This has now been corrected in the manuscript.

7. Line 237: “The separation was greater for scaled data showing that smaller peaks were contributing to the difference in PS80 Type 2 samples.”

It is interesting if you are assessing the peaks of each type to be able to see these peaks.

As described in point (5) above, this section of the results had some mistakes, and has now been edited corrected.

8. Line 242 . “Type 2 had the lowest degree of ethoxylation with 2 to 3 EO units less than the other types.”

You said that type 2 is the one that shows the lower degree of ethoxylation, but is it really type 1 that differs. Is it really type 1 that shows the lower degree of ethoxylation? Or is it type 2, and the separation from type 1 is due to another factor that needs to be discussed?

As described in point (5) above, this section of the results had some mistakes, and has now been edited corrected.

9. Line 271: “This difference can also be seen in partial least squares (PLS) analysis of the extracted distribution features.”

Where? If it is figure 6 please quote it in that sentence.

This has now been made clearer in the manuscript, as suggested by the reviewer.

10. As a general comment, much emphasis is made in the paper that this methodology proposed here is a useful tool for people without a deep knowledge of mass spectrometry. I understand the message, that the aim is to provide a tool that is easy to apply and versatile in the industry, which can be handled by any personnel. On the other hand, I see it unnecessary to highlight the lack of knowledge of mass spectrometry as a quality of the methodology. I would comment more on its simplicity and ease of application. I think it is possible to highlight the versatility of the methodology without the need to claim the lack of knowledge in mass spectrometry, which was necessary to create this versatile methodology. I believe that this approach is more commercially oriented than academic.

We thank the reviewer for their positive comments. We have now highlighted the versatility and simplicity of the method in the manuscript, as the reviewer suggested.

11. On the other hand, although I understand that it is a pioneering study in this area, the paper itself has only 16 bibliographical references, 11 of which are in the first 12 lines of the introduction. There is very little comparison of the results obtained with any other study, and there is very little discussion with other approaches. On the other hand, the introduction is a bit short, as it gives a quick state of the art, but does not mention ethoxylates to any great extent. Thus, the chosen types polysorbate, PEG and oleth-20 are not mentioned. Applications, degree of use, general interest...

Extra detail has now been included in the introduction and further bibliographic references as suggested by the reviewer. More details have been included on the chosen types of studied ethoxylates.

12. On the other hand, I congratulate the authors, the approach is attractive, and I hope it will be a tool that can be used in a common way.

We thank the reviewer for their positive comments.

---

## [Decision Letter · Decision Letter 1]

PONE-D-24-58607R1Chemometric analysis of ethoxylated polymer products using extracted MALDI-TOF-MS peak distribution features.PLOS ONE

Dear Dr. Wilson,

Thank you for submitting your manuscript to PLOS ONE. After careful consideration, we feel that it has merit but does not fully meet PLOS ONE’s publication criteria as it currently stands. Therefore, we invite you to submit a revised version of the manuscript that addresses the points raised during the review process.

**ACADEMIC EDITOR: Minor revision**==============================

We look forward to receiving your revised manuscript.

Kind regards,

Muhammad Atif

Academic Editor

PLOS ONE

Journal Requirements:

Additional Editor Comments:

Thank you for submitting your manuscript. After reviewing it, I believe that it has great potential. However, I do require some minor revisions before it can be accepted for publication. Please address the specific comments provided and resubmit your revised manuscript.

Reviewers' comments:

Reviewer's Responses to Questions

**Comments to the Author**

1. If the authors have adequately addressed your comments raised in a previous round of review and you feel that this manuscript is now acceptable for publication, you may indicate that here to bypass the “Comments to the Author” section, enter your conflict of interest statement in the “Confidential to Editor” section, and submit your "Accept" recommendation.

Reviewer #2: All comments have been addressed

Reviewer #3: All comments have been addressed

2. Is the manuscript technically sound, and do the data support the conclusions?

Reviewer #2: Yes

Reviewer #3: Yes

3. Has the statistical analysis been performed appropriately and rigorously? 

Reviewer #2: Yes

Reviewer #3: Yes

4. Have the authors made all data underlying the findings in their manuscript fully available?

Reviewer #2: Yes

Reviewer #3: Yes

5. Is the manuscript presented in an intelligible fashion and written in standard English?

Reviewer #2: Yes

Reviewer #3: Yes

6. Review Comments to the Author

Reviewer #2: (No Response)

Reviewer #3: 1) In Table 1, I appreciate that my comment has been taken into account. However, part of the idea of unifying the content is to simplify it. In the first column, Product, I believe that simply centring the name of each product (Polysorbate 80, PEG Castor Oil, Oleth-20) would be sufficient.

2) In relation to my tenth comment, while the practicality of the methodology described has been successfully highlighted, my observation emphasised that the paper underscores in the abstract (L43):

"As expertise in mass spectrometry is not required to interpret the results."

And in the conclusion (L374):

"Most importantly, our method provides results that are easily interpreted, and can be understood without the need for mass-spectrometry expertise."

This emphasis conveys a more commercial rather than an academic approach. I understand that your methodology simplifies the analysis and allows individuals without extensive knowledge of mass spectrometry to use it. However, I believe that this feature could have been highlighted without explicitly stressing the lack of necessity for mass spectrometry expertise, particularly in a study that focuses on a chemometric analysis incorporating spectrometric detectors.

At the same time, I consider these to be minor corrections, as the paper has been successfully revised through the input of various reviewers, which I believe has significantly enhanced its overall quality.

7. PLOS authors have the option to publish the peer review history of their article (what does this mean? ). If published, this will include your full peer review and any attached files.

**Do you want your identity to be public for this peer review?** For information about this choice, including consent withdrawal, please see our Privacy Policy .

Reviewer #2: No

Reviewer #3: **Yes: ** Aly Castillo

---

## [Author Response · Author response to Decision Letter 2]

10 Apr 2025

In response to Reviewer 3's comments, we have now simplified Table 1 by merging cells and only providing the

name of each product once. For the second comment, we apologise for misunderstanding the original comment. We appreciate

that the lack of mass spectrometry knowledge required to interpret the results is more relevant to industry than academia. In the abstract,

the sentence "As expertise in mass spectrometry is not required to interpret the results, the method has the potential

to aid comparison of closely related products, for example, made at different geographical locations or in

different reactors, therefore supporting quality assurance and improving ‘right first time’ rates."

has been replace with

"This simplification facilitates interpretation of the results and allows the comparison of closely related products."

However, we do think that the relevance to industry is important and so have left the sentence

"This makes the results accessible to non-MS specialists and provides a methodology suitable for comparison in

industrial polymer manufacture. Such comparisons are essential for quality assurance to ensure that product composition

remains consistent for customers, perhaps when the same product may be manufactured at different global locations and

is reliant on naturally varying raw materials, or perhaps when manufacture is scaled up or moved between different

industrial plants." in the introduction.

In the conclusions, we have replaced

"Most importantly, our method provides results that are easily interpreted, and can be understood without the need for mass-spectrometry expertise."

with

"Most importantly, our method simplifies analysis by providing results that are easily interpreted."

---

## [Decision Letter · Decision Letter 2]

PONE-D-24-58607R2Chemometric analysis of ethoxylated polymer products using extracted MALDI-TOF-MS peak distribution features.PLOS ONE

Dear Dr. Wilson,

Thank you for submitting your manuscript to PLOS ONE. After careful consideration, we feel that it has merit but does not fully meet PLOS ONE’s publication criteria as it currently stands. Therefore, we invite you to submit a revised version of the manuscript that addresses the points raised during the review process.

Your manuscript has been evaluated by Reviewers 2 and 3 from the previous round of peer review. Both reviewers are satisfied with your revisions and recommend publication. Before we can proceed, we note that your manuscript reports the use of C and R code in the methodology. Please note that PLOS ONE has specific guidelines on code sharing for submissions in which author-generated code underpins the findings in the manuscript. In these cases, we expect all author-generated code to be made available without restrictions upon publication of the work. Please review our guidelines at https://journals.plos.org/plosone/s/materials-and-software-sharing#loc-sharing-code and ensure that your code is shared in a way that follows best practice and facilitates reproducibility and reuse. No other revisions are required, and I do not anticipate that further review will be necessary.

We look forward to receiving your revised manuscript.

Kind regards,

Hugh Cowley

Senior Editor

PLOS ONE

Journal Requirements:

Reviewers' comments:

Reviewer's Responses to Questions

**Comments to the Author**

1. If the authors have adequately addressed your comments raised in a previous round of review and you feel that this manuscript is now acceptable for publication, you may indicate that here to bypass the “Comments to the Author” section, enter your conflict of interest statement in the “Confidential to Editor” section, and submit your "Accept" recommendation.

Reviewer #2: (No Response)

Reviewer #3: All comments have been addressed

2. Is the manuscript technically sound, and do the data support the conclusions?

Reviewer #2: (No Response)

Reviewer #3: Yes

3. Has the statistical analysis been performed appropriately and rigorously? 

Reviewer #2: (No Response)

Reviewer #3: Yes

4. Have the authors made all data underlying the findings in their manuscript fully available?

Reviewer #2: (No Response)

Reviewer #3: Yes

5. Is the manuscript presented in an intelligible fashion and written in standard English?

Reviewer #2: (No Response)

Reviewer #3: Yes

6. Review Comments to the Author

Reviewer #2: (No Response)

Reviewer #3: The paper “Chemometric analysis of ethoxylated polymer products using extracted

MALDI-TOF-MS peak distribution features”, is a novel study that encompasses a new

approach for the differentiation of ethoxylated products by means of an algorithm

applied to the data obtained from the analysis of ethoxylated products by MALDI-TOF-

MS. Thus, the work combines a large number of models for the treatment of mass

spectral data, a deep linkage of high-resolution MALDI-TOF-MS analysis generating a

versatile tool suitable for all users. All comments have been answered. I recommend posting as current state.

7. PLOS authors have the option to publish the peer review history of their article (what does this mean? ). If published, this will include your full peer review and any attached files.

**Do you want your identity to be public for this peer review?** For information about this choice, including consent withdrawal, please see our Privacy Policy .

Reviewer #2: No

Reviewer #3: **Yes: ** Aly Castillo

---

## [Author Response · Author response to Decision Letter 3]

25 Jun 2025

As the reviewers are satisfied with our revisions and recommend publication as is, we have made no further changes to the manuscript, so that the file labeled 'Revised Manuscript with Track Changes' is now the same as the file labeled 'Manuscript'.

We have now made our C and R code available together with all data files via the Open Science Framework (OSF) database (DOI 10.17605/OSF.IO/M46T5) and the data availability statement now reads "All data files together with C and R code for data processing are available from the Open Science Framework (OSF) database (DOI 10.17605/OSF.IO/M46T5)."

---

## [Editor Report · Decision Letter 3]

Chemometric analysis of ethoxylated polymer products using extracted MALDI-TOF-MS peak distribution features.

PONE-D-24-58607R3

Dear Dr. Wilson,

We’re pleased to inform you that your manuscript has been judged scientifically suitable for publication and will be formally accepted for publication once it meets all outstanding technical requirements.

Kind regards,

Hugh Cowley

Staff Editor

PLOS ONE
---

## [Editor Report · Acceptance letter]

PONE-D-24-58607R3

PLOS ONE

Dear Dr. Wilson,

I'm pleased to inform you that your manuscript has been deemed suitable for publication in PLOS ONE. Congratulations! Your manuscript is now being handed over to our production team.

Kind regards,

on behalf of

Mr Hugh Cowley

Staff Editor

PLOS ONE